

# Improving the physical, mechanical and energetic properties of *Quercus* spp. wood pellets by adding pine sawdust

Víctor Daniel Núñez-Retana[1], Rigoberto Rosales-Serna[2], José Ángel Prieto-Ruíz[3], Christian Wehenkel[4] and Artemio Carrillo-Parra[4]

[1] Maestría Institucional en Ciencias Agropecuarias y Forestales, Universidad Juárez del Estado de Durango, Durango, Durango, Mexico
[2] Campus Valle del Guadiana, Instituto Nacional de Investigaciones Forestales, Agrícolas y Pecuarias, Durango, Durango, México
[3] Facultad de Ciencias Forestales, Universidad Juárez del Estado de Durango, Durango, Durango, Mexico
[4] Instituto de Silvicultura e Industria de la Madera, Universidad Juárez del Estado de Durango, Durango, Durango, Mexico

Corresponding author
Artemio Carrillo-Parra,
acarrilloparra@ujed.mx

## ABSTRACT

**Background:** Biomass usage for energy purposes has emerged in response to global energy demands and environmental problems. The large amounts of by-products generated during logging are rarely utilized. In addition, some species (e.g., *Quercus* spp.) are considered less valuable and are left in the cutting areas. Production of pellets from this alternative source of biomass may be possible for power generation. Although the pellets may be of lower quality than other types of wood pellets, because of their physical and technological properties, the addition of different raw materials may improve the characteristics of the oak pellets.

**Methods:** Sawdust from the oak species *Quercus sideroxyla, Q. rugosa, Q. laeta* and *Q. conzattii* was mixed with sawdust from the pine *Pinus durangensis* in different ratios of oak to pine (100:0, 80:20, 60:40, 40:60 and 20:80). Physical and mechanical properties of the pellets were determined, and calorific value tests were carried out. For each variable, Kolmogorov–Smirnov normality and Kruskal–Wallis tests were performed and Pearson's correlation coefficients were determined (considering a significance level of $p < 0.05$).

**Results:** The moisture content and fixed carbon content differed significantly ($p < 0.05$) between the groups of pellets (i.e., pellets made with different sawdust mixtures). The moisture content of all pellets was less than 10%. However, volatile matter and ash content did not differ significantly between groups ($p \geq 0.05$). The ash content was less than 0.7% in all mixtures. The addition of *P. durangensis* sawdust to the mixtures improved the bulk density of the pellets by 18%. Significant differences ($p < 0.05$) in particle density were observed between species, mixtures and for the species × mixture interaction. The particle density was highest in the 80:20 and 60:40 mixtures, with values ranging from 1,245 to 1,349 kg m$^{-3}$. Bulk density and particle density of the pellets were positively correlated with the amount of *P. durangensis* sawdust included. The mechanical hardness and impact resistance index (IRI) differed significantly ($p < 0.05$) between groups. The addition of pine sawdust decreased the mechanical hardness of the pellets, up to 24%. The IRI was

highest (138) in the *Q. sideroxyla* pellets (100:0). The mechanical hardness and IRI of the pellets were negatively correlated with the amount of *P. durangensis* sawdust added. The bulk density of the pellets was negatively correlated with mechanical hardness and IRI. The calorific value of mixtures and the species × mixture interaction differed significantly between groups. Finally, the mean calorific value was highest (19.8 MJ kg$^{-1}$) in the 20:80 mixture. The calorific value was positively related to the addition of *P. durangensis* sawdust.

# INTRODUCTION

The increase in the world's population has created a greater demand for fossil fuels, which has led to a scarcity of these materials and to unstable prices. According to the United Nations Population Fund (*UNFPA, 2019*) the world's population was 7,000 million in 2010 and close to 7,715 million in 2019, representing an average annual rate of population change of 1.1% (for the period 2010–2019). Plant biomass has become an important renewable resource and currently covers approximately 15% of total energy consumption in the world (*Holubcik, Jandacka & Durcansky, 2016*).

Mexican temperate forests are dominated by pine-oak species (*Galicia, Potvin & Messier, 2015*). Mexican pine-oak forests, which cover an area of 31.8 million hectares (*FAO, 1998*), are commonly uneven-aged mixed forests (*Wehenkel et al., 2011*; *Maciel-Nájera et al., 2020*). *Pinus* wood production reached 5.0 million of m$^3$ of roundwood in the last decade (*SEMARNAT, 2016*). Although *Quercus* spp. represent the second most important Mexican forest timber resource, covering an area of about 8.4 ha and yielding annual wood production of about 738,000 m$^3$ of roundwood (*SEMARNAT, 2016*), these species remain almost underutilized (*Bárcenas-Pazos et al., 2008*; *Villela-Suárez et al., 2018*).

Current methods of forest harvesting usually select pine species for harvesting, and logging activities also generate large amounts of by-products in the form of tree branches, tips, bark and sawdust. Logging thus changes the forest structure and species composition promoting the dominance of some species of low economic value, such as Mexican oaks in pine-oak forests (*Moreno-Lopez, Alarcón-Herrera & Martin-Dominguez, 2017*). The dominance of oak trees interferes with natural restoration of pine populations under intensive wood production in temperate forests. The presence of some *Quercus* species has been associated with negative effects such as shading (*Puértolas, Benito & Peñuelas, 2009*), allelopathy, restrained seed germination and seedling radicle growth and inhibition of nitrifying bacteria, thus affecting the self-restoration of ponderosa pine and the herb understory (*Li, Jia & Li, 2007*).

In Mexico, the disposal of solid timber by-products can create problems in forestlands and sawmills as it can lead to forest fires during periods of intense heat, generate dust in the air and block spaces in production installations (*Fregoso-Madueño et al., 2017*). Furthermore, pine wood is destined for the production of firewood, pulp, resin, edible

seeds and other products such as furniture and boards (*Sánchez, 2008*) and therefore it should not be used to produce bioenergy. By-products and poorly formed stems and mature wood from oak trees could be used as an alternative source of material to produce bioenergy. Nevertheless, oak material is rarely transformed into pellets, because of technical problems due to the anatomical, physical, mechanical and drying characteristics of the timber (*Miranda et al., 2011*). The density of oak wood is considered medium to high (401–800 kg m$^{-3}$) (*Herrera-Fernández et al., 2017*), which may lead to machining problems in sawmill systems during conversion (*Zavala Zavala, 2003*; *Herrera-Fernández et al., 2017*).

Pressing biomass into pellets has emerged as an efficient means of creating a renewable energy resource. However, not all species are easily pelletized, and the quality of the pellets is determined by the physical, mechanical, chemical and energetic properties. Mechanical properties such as strength and durability can be measured by compressive resistance testing, the tumbling can method, the Holmen Ligno tester and by impact resistance and water resistance methods (*Kaliyan & Morey, 2009*).

When the physical, mechanical and energetic properties of the pellets do not reach international standards, the quality can be improved by the use of mixtures of material to make the pellets. Indeed, researchers such as *Kaliyan & Morey (2009)* and *Harun & Afzal (2016)* recommend using mixtures of raw materials. Thus, *Wilson (2010)* mixed pine sawdust with white oak and red oak sawdust, thereby improving the durability of the pellets. *Miranda et al. (2009)* showed that pellets made from *Quercus pyrenaica* residues were suitable for energy applications. The same researcher used mixtures of Pyrenean oak and washed grape pomace to make pellets, which proved to have good physical and thermal properties (*Miranda et al., 2011*). *Arranz et al. (2015)* compared commercial pellets and an experimental type of Pyrenean oak pellet made in a semi-industrial pelletizer and found that the calorific values produced by some pellets were sufficient. These researchers therefore recommend taking specific actions to improve the pellet quality and optimize the operations in relation to collecting and handling the raw material. Similarly, *Monedero, Portero & Lapuerta (2015)* recommended the addition of pine sawdust to poplar chips (*Populus* spp.) before pelletizing to improve the pellet quality and enable compliance with the established requirements of the standard EN 14961-2 (*Spanish Association for Standarization (UNE), 2012*).

The aim of this study was to improve the physical, mechanical and energetic properties of oak wood pellets (without bark) by mixing the oak sawdust with pine (*P. durangensis*) sawdust in different proportions before pelletizing the material.

## MATERIALS AND METHODS

### Raw materials and experimental design

Specimens of the oak species *Quercus sideroxyla*, *Q. rugosa* and *Q. laeta* were collected in the cutting areas in the Llano Blanco (SG.FO-08/-2014/91), El Nopal (SG.FO-08-2014/129), Chinatú (SG.FO-08-2014/52), El Pinito (SG.FO-08/2015/40) and El Tule y Portugal (SG.FO-08-2014/82) forest communities located in the municipality of Guadalupe y Calvo, state of Chihuahua, Mexico. Specimens of another oak species, *Quercus conzattii*,

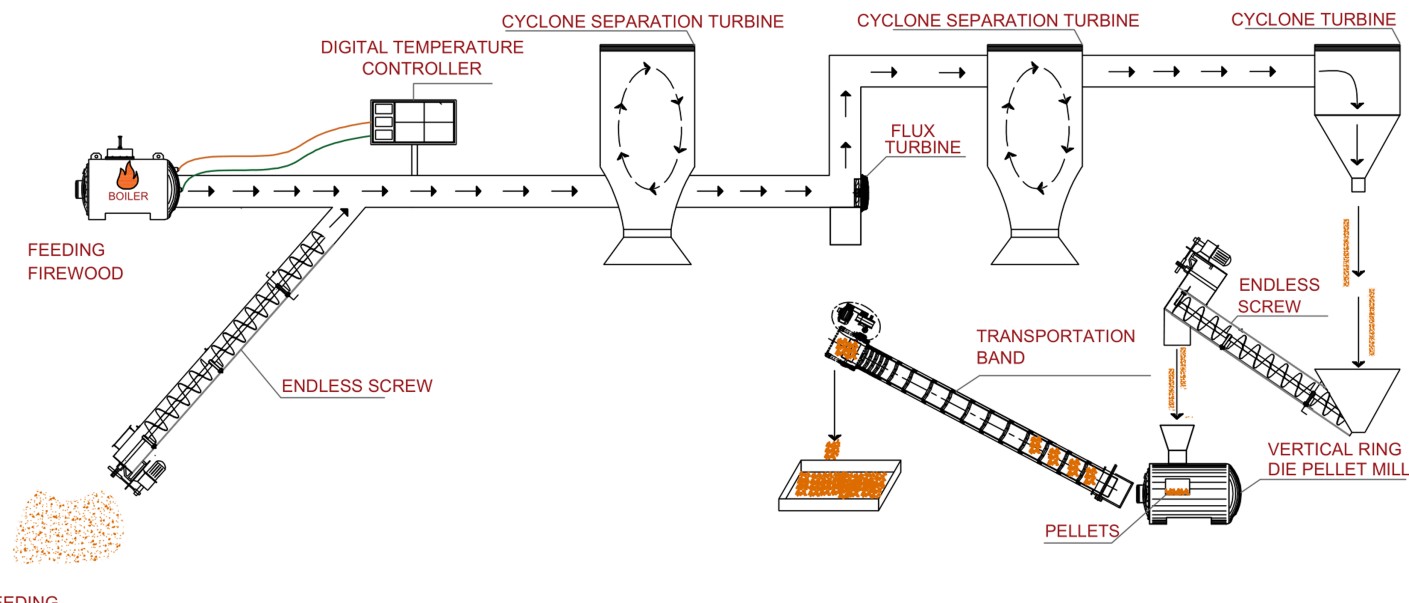

**Figure 1** Schematic representation of the pelletizing process used in the Instituto de Silvicultura e Industria de la Madera of the Universidad Juárez del Estado de Durango (made by Víctor Daniel Núñez-Retana).

were collected from the Nicolás Romero forest area (SG/130.2.2.2/002203/17), and *Pinus durangensis* specimens were obtained from the El Regocijo forest community (SG/130.2.2.2/002243/11), both in the municipality of Durango, state of Durango, Mexico.

The material was collected by motor-manual harvesting, as follows: four logs were cut from pest- and disease-free specimens of each species, of diameter at breast height (dbh) ≥ 25 cm; the stem was required to be straight at least until the proposed height for cutting (1.30 m) due to the complexity for chipping.

## Preparation of biomass raw material and pellet manufacturing

The logs were seasoned under laboratory conditions and debarked, cut and chipped using an Industrial Duty (SD4P25T61Y) machine. The sawdust was produced in a hammer mill (TFS 420) with a 3.15 mm mesh. The sawdust from each oak species was mixed with pine sawdust without bark in the following proportions (oak:pine): 100:0, 80:20, 60:40, 40:60 and 20:80. Ten kg of each mixture was prepared for pellet manufacturing. The sawdust was placed in rubber bags and mixed homogeneously.

The sawdust was conditioned by controlling the temperature of the boiler, by means of a digital controller, until 10% humidity was reached. The sawdust was mechanically transported to the entrance of the ZLSP-R300 pelletizer to form the pellets (Fig. 1).

The pelletizer consists of a flat disc with channels 8 mm long and 6 mm wide and produces pellets at a rate of 400 kg h$^{-1}$. Before the samples were pelleted, the temperature of the pelletizer was increased by processing pine sawdust only. The pelletizer was then constantly fed with the sawdust mixture until 8 kg of material per mixture was formed. The pellets were cooled by holding at room temperature for 24 h (Fig. 2).
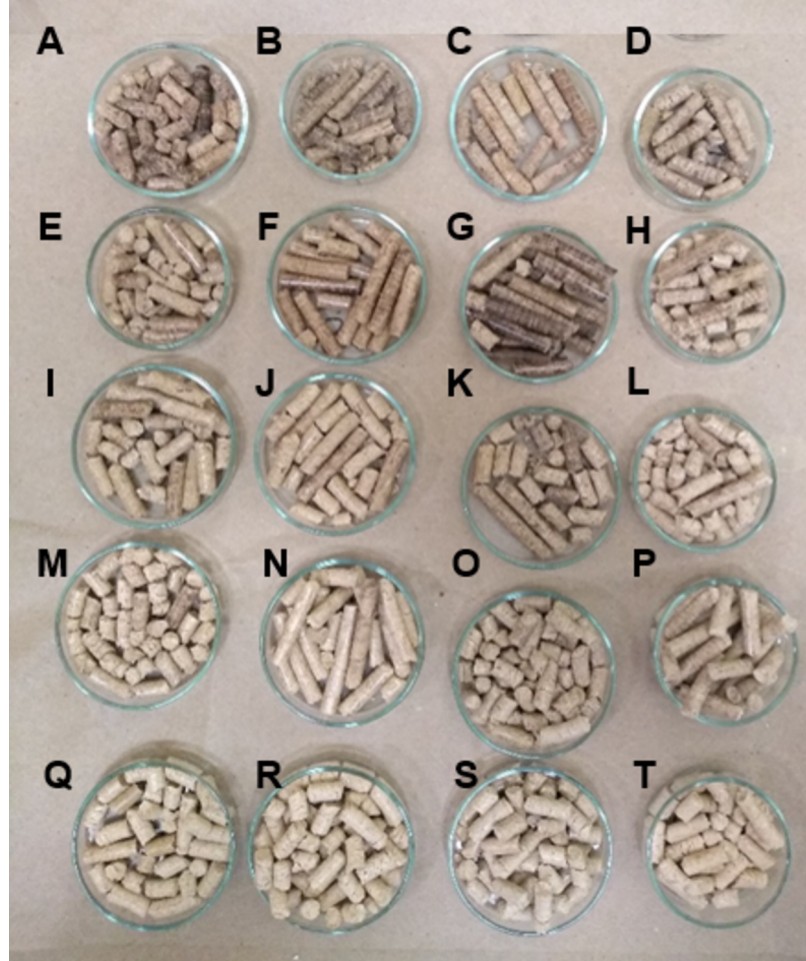

**Figure 2 Pellets of different mixtures of oak-pine sawdust.** Horizontally (A–D) Mixtures of 100:0. (E–H) Mixtures of 80:20. (I–L) Mixtures of 60:40. (M–P) Mixtures of 40:60. (Q–T) Mixtures of 20:80. Vertically (A–Q) *Q. sideroxyla.* (B–R) *Q. rugosa.* (C–S) *Q. laeta.* (D–T) *Q. conzattii* species.

## Proximate analysis

The cooled pellets were subjected to proximate analysis. The moisture content of the pellets was determined according to EN 18134-3 (*Spanish Association for Standarization (UNE), 2016a*). Samples were weighed on a 1 mg precision weighing scale before and after drying in an oven for 4 h at 105 ± 2 °C. The volatile matter was measured following standard EN 18123 (*Spanish Association for Standarization (UNE), 2016b*) in which the samples are heated at 900 ± 10 °C for 7 min.

The ash content was measured according to standard EN 18122 (*Spanish Association for Standarization (UNE), 2016c*). Thus, the samples were initially weighed and placed in a muffle at 250 °C for 1 h, and the temperature was then increased to 550 °C for 2 h. The final weight was determined after cooling the samples in a desiccator. The amount of fixed carbon was calculated by subtracting the sum of moisture content, volatile matter and ash from 100% (*Carrillo-Parra et al., 2018*).

## Physical properties

Bulk density tests were carried out in triplicate in a 600 mL metal cylinder, according to the procedure outlined in standard EN-17828 (*Spanish Association for Standarization (UNE), 2016d*), in which the pellets were poured into the cylinder until it was full. A debris cone was then formed. The cylinder was then struck three times on a hard surface from a height of 150 mm to consolidate the pellets, and excess pellets were removed from the edge of the cylinder.

The pellet particle density (kg m$^{-3}$) was estimated by measuring the weight and volume of 20 pellets.

## Mechanical properties

The mechanical hardness was estimated by means of the drop test. The test consists of measuring the weight of each pellet before and after being dropped twice from a height of 1.85 m onto a concrete floor. Twenty repetitions were carried out per treatment. The impact resistance index (IRI) was then calculated as described by *Richards (1990)*, (IRI) = $(100 \times N)/n$, where $N$ is the number of drops, and $n$ is the total number of pieces after $N$ drops. The maximum IRI value is 200. Small pieces weighing less than 5% of the total pellet weight were not considered.

## Energetic properties

The pellet calorific value was calculated in a semi-automatic isoperibol calorimeter (LECO model AC600) in TruSpeed® mode and according to standard EN-14918 (*Spanish Association for Standarization (UNE), 2011a*). The sample for analysis was burned with a high oxygen pressure in a calorimetric pump under specified conditions. The tests were carried out in triplicate on an anhydrous basis. The calculation was performed automatically by the calorimeter.

## Statistical analysis

Kolmogorov–Smirnov normality tests and analysis of variance were performed for all the variables according the assumption of normality. Statistical analysis of the bulk density, particle density, calorific value, mechanical hardness and IRI data were performed according to a factorial design ($4 \times 5$), for the factors *species* (4) and *mixture* (5). A Kruskal–Wallis test was applied for non-normally distributed variables. Pearson's correlation coefficients were calculated in order to evaluate the strength of association between the addition of *Pinus durangensis* sawdust and bulk density, mechanical hardness, IRI and calorific value, as well as between bulk density and mechanical hardness and IRI. All tests were performed considering a significance level of $p < 0.05$ and were implemented in the statistical program RStudio® version 3.2.2 R (*Bolker, 2012*).

# RESULTS

## Proximate analysis

The moisture content differed significantly between species ($p = 4.09 \times 10^{-5}$), mixtures ($p = 1.14 \times 10^{-7}$) and for the species × mixture interaction ($p = 0.001$) (Table 1).

**Table 1 Proximal analysis of pellets made from different mixtures of oak and *P. durangensis* sawdust.**

| Factor | MC (%) | MV (%) | AC (%) | FC (%) |
|---|---|---|---|---|
| Species | | | | |
| *Q. sideroxyla* | 2.14 C | 90.64 A | 0.54 | 6.65 B |
| *Q. rugosa* | 3.22 A | 87.33 C | 0.55 | 8.88 A |
| *Q. laeta* | 2.90 B | 89.63 A | 0.51 | 6.93 B |
| *Q. conzattii* | 2.35 B | 88.64 B | 0.57 | 8.44 A |
| *P. durangensis** | 4.80 | 80.68 | 0.53 | 13.91 |
| Mixture | | | | |
| 100:0 | 3.78 A | 87.54 | 0.53 | 8.13 |
| 80:20 | 2.09 B | 88.98 | 0.51 | 8.40 |
| 60:40 | 2.24 B | 89.70 | 0.55 | 7.48 |
| 40:60 | 2.35 B | 89.99 | 0.55 | 7.09 |
| 20:80 | 2.78 B | 89.11 | 0.57 | 7.51 |
| *Q. sideroxyla–P. durangensis* | | | | |
| 100:0 | 3.71 b | 89.76 | 0.36 | 6.15 k |
| 80:20 | 1.10 m | 91.34 | 0.47 | 7.07 j |
| 60:40 | 1.21 l | 90.82 | 0.60 | 7.36 i |
| 40:60 | 2.11 j | 91.08 | 0.65 | 6.13 k |
| 20:80 | 2.57 h | 90.23 | 0.62 | 6.56 k |
| *Q. rugosa–P. durangensis* | | | | |
| 100:0 | 3.36 e | 84.00 | 0.74 | 11.88 a |
| 80:20 | 3.28 f | 87.48 | 0.46 | 8.76 d |
| 60:40 | 3.38 e | 87.99 | 0.53 | 8.08 e |
| 40:60 | 3.43 d | 88.01 | 0.42 | 8.12 e |
| 20:80 | 2.65 g | 89.18 | 0.57 | 7.58 g |
| *Q. laeta–P. durangensis* | | | | |
| 100:0 | 4.00 a | 88.45 | 0.52 | 7.01 j |
| 80:20 | 2.64 g | 88.67 | 0.51 | 8.15 e |
| 60:40 | 2.66 g | 89.91 | 0.48 | 6.92 j |
| 40:60 | 1.72 k | 90.83 | 0.55 | 6.89 j |
| 20:80 | 3.50 c | 90.31 | 0.51 | 5.66 l |
| *Q. conzattii–P. durangensis* | | | | |
| 100:0 | 4.05 a | 87.95 | 0.49 | 7.49 h |
| 80:20 | 1.35 k | 88.42 | 0.59 | 9.62 c |
| 60:40 | 1.73 k | 90.07 | 0.60 | 7.58 f |
| 40:60 | 2.14 j | 90.03 | 0.58 | 7.23 i |
| 20:80 | 2.42 i | 86.72 | 0.58 | 10.26 b |

**Notes:**
* Pine values are added as a comparison parameter.
MC, Moisture content; VM, Volatile matter; AC, Ash content; FC, Fixed carbon. Different letters correspond to significant statistical differences $p < 0.05$. Capital letters corresponds to species and mixtures; lowercase letters correspond to species × mixture interaction.

The moisture content was lowest in the pellets made from *Q. sideroxyla* (2.0%), followed by those made from *Q. conzattii* (2.3%) and *Q. laeta* (2.7%) and *Q. rugosa* (3.3%). The moisture content was highest in the pellets made from oak sawdust only (for all species). The pellets made from *Q. sideroxyla* mixed with pine in ratios of 80:20 and 60:40 had the lowest moisture contents, of 1.1% and 1.2%, respectively. The moisture content of *P. durangensis* pellets (4.8%) was higher than that of the pellets made from any of the mixtures.

The volatile matter content differed significantly between species ($p = 0.001$), but not between mixtures or for the species × mixture interaction ($p = 0.07$ and $0.63$, respectively) (Table 1). The mean values were highest in *Q. sideroxyla* and *Q. laeta* pellets (90.6% and 89.6%, respectively), followed by *Q. conzattii* (88.6%) and *Q. rugosa* (87.3%) pellets. The values were in the range 81–91%, and the volatile matter content of the pine pellets (80.6%) was lower than in the pellets made from any of the mixtures.

There were no significant differences in ash content between species ($p = 0.83$), mixtures ($p = 0.86$) or for the species × mixture interaction ($p = 0.30$) (Table 1). However, in all mixtures including *P. durangensis*, the ash content was below 0.7% (except *Q. rugosa* 100–0).

Fixed carbon differed significantly between species ($p = 0.002$) and for the species × mixture interaction ($p = 0.001$), while there were no significant differences for mixtures ($p = 0.40$) (Table 1). The mean fixed carbon content was highest in *Q. rugosa* (8.8%) and *Q. conzattii* (8.4%) followed by *Q. laeta* (6.9%) and *Q. sideroxyla* (6.6%) pellets. On the other hand, the fixed carbon content was highest in *Q. rugosa* pellets (100:0) (11.8%) and lowest in the *Q. laeta*: *P. durangensis* 20:80 mixture (5.6%). The fixed carbon content of *P. durangensis* pellets (13.9%) was higher than in all mixtures.

## Physical properties

Bulk density varied in the range 557–703 kg m$^{-3}$. The bulk density did not vary significantly between species ($p = 0.18$) or for the species × mixture interaction ($p = 0.99$), but it did differ significantly between the mixtures ($p = 1.04 \times 10^{-6}$) (Fig. 3A). The value was highest in all 20:80 mixtures (>646 kg m$^{-3}$) and lowest in the oak-only pellets (100:0) (<580 kg m$^{-3}$). The bulk density of the *P. durangensis* pellets was 647 kg m$^{-3}$.

The particle density of pellets differed significantly between species ($p = 0.01$), mixtures ($p = 1.08 \times 10^{-8}$) and for the species × mixture interaction ($p = 1.49 \times 10^{-11}$) (Fig. 3B). The mean particle density was highest in *Q. laeta* (1,282 kg m$^{-3}$) followed by *Q. sideroxyla* (1,257 kg m$^{-3}$) and *Q. rugosa* (1,256 kg m$^{-3}$) and *Q. conzattii* (1,246 kg m$^{-3}$) pellets. The particle density was highest in the 80:20 and 60:40 mixtures and varied in the range 1,245–1,349 kg m$^{-3}$. The particle density was highest in the 80:20 mixture of *Q. laeta* and *P. durangensis*. The particle density of the *P. durangensis* pellets was 1,227 kg m$^{-3}$.

The pellet bulk density was positively correlated with the amount of *P. durangensis* sawdust added (Fig. 3A). The bulk density of *Q. laeta* was most closely correlated ($r = 0.82$) with the amount of *P. durangensis* sawdust added, while that of *Q. conzattii* was least well correlated ($r = 0.71$). On the other hand, except for *Q. rugosa*, the particle density was poorly correlated with the amount of *P. durangensis* sawdust added (Fig. 3B).

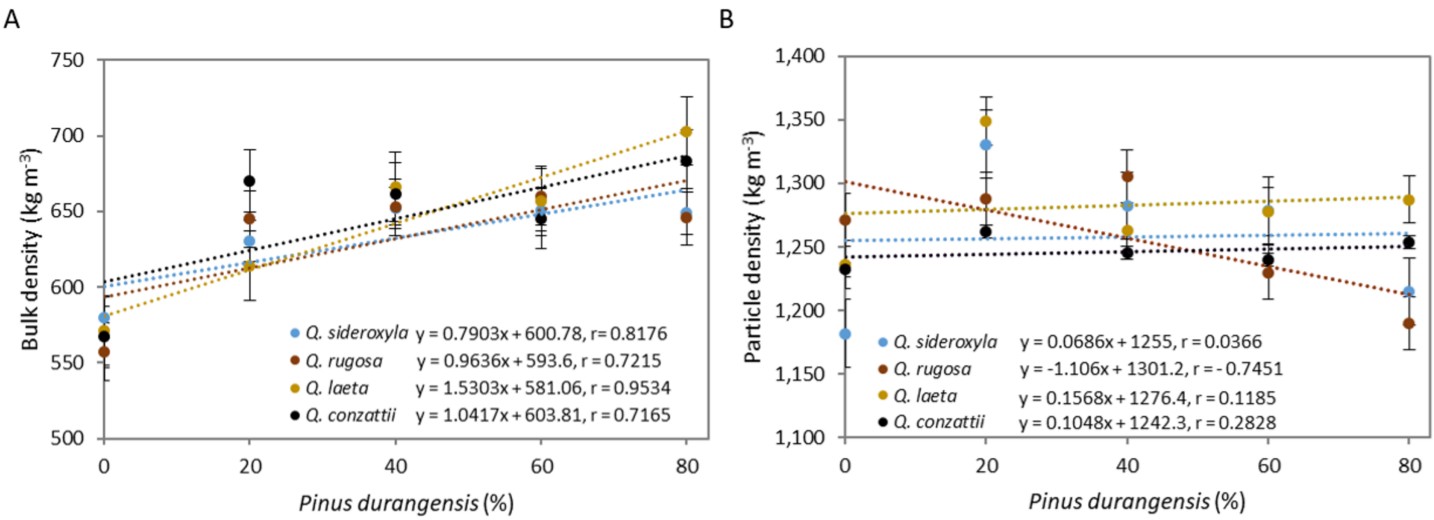

**Figure 3** Addition of *P. durangensis* sawdust correlated with pellets bulk density (A) and particle density (B) of four oak species.

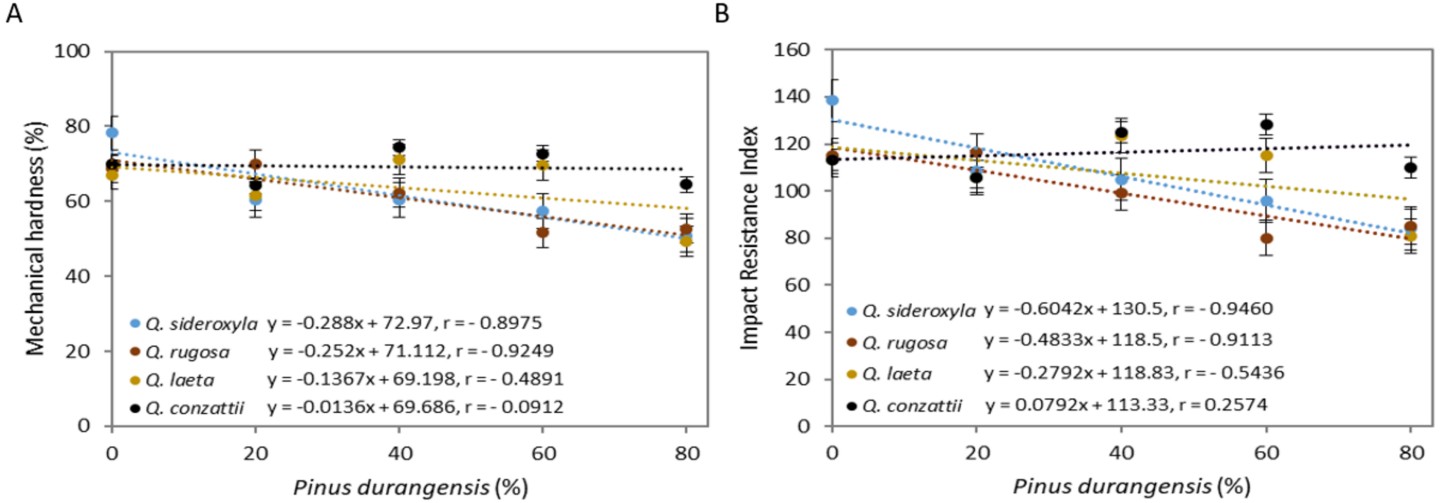

**Figure 4** Addition of *P. durangensis* sawdust correlated with pellets mechanical hardness (A) and Impact Resistance Index (B) of four oak species.

## Mechanical properties

The mechanical hardness differed significantly between species ($p = 0.01$), mixtures ($p = 7.32 \times 10^{-7}$) and for the species × mixture interaction ($p = 0.001$) (Fig. 4A). The mean percentage of retained mass was highest in *Q. conzattii* (69.1%), followed by *Q. laeta, Q. sideroxyla* and *Q. rugosa* pellets: 63.7%, 61.4% and 61.0%, respectively.

The mean percentage of retained mass was highest in all oak-only pellets (100:0) (71%) and lowest in the 20:80 mixtures (54.3%). The percentage of retained mass was highest in the *Q. sideroxyla*-only pellets (100:0) (78.2%). The mean retained mass in the *P. durangensis* pellets was 58.4%.

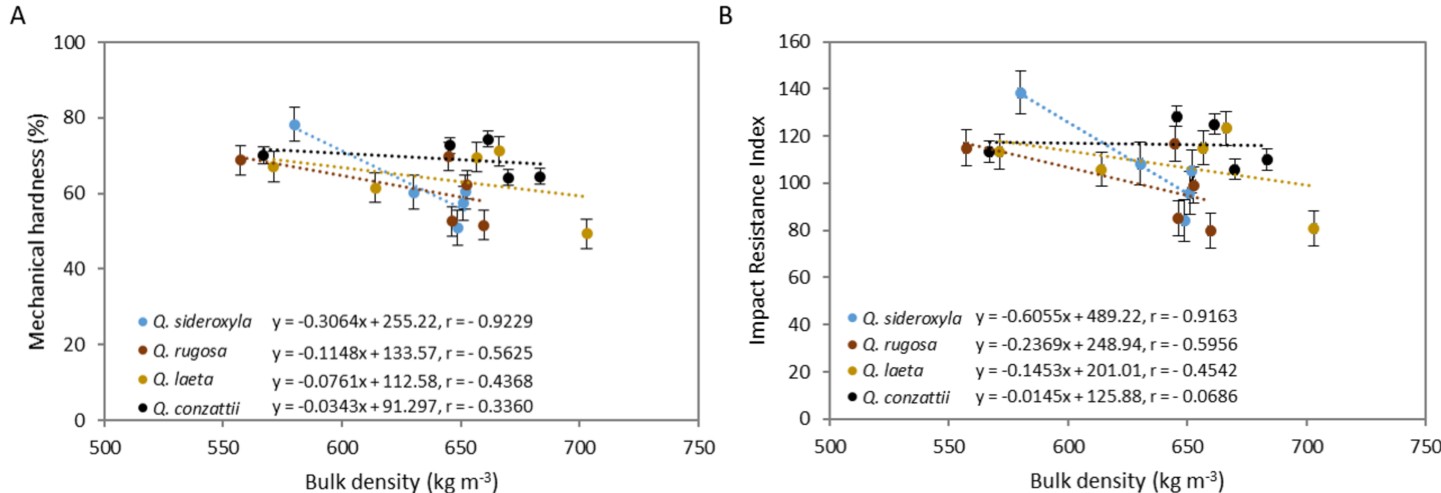

**Figure 5 Pellets bulk density correlated with mechanical hardness (A) and Impact Resistance Index (B) of four oak species.**

The IRI values also differed significantly between species ($p = 0.01$), mixtures ($p = 3.64 \times 10^{-6}$) and for the species × mixture interaction ($p = 0.01$) (Fig. 4B). The mean value was highest in *Q. conzattii* (116), followed by *Q. laeta* (107), *Q. sideroxyla* (106) and *Q. rugosa* (99) pellets. The mean IRI value was highest in the oak-only pellets (100:0) (120) and lowest in the 20:80 mixtures. On the other hand, the value was highest in the *Q. sideroxyla*-only pellets (100:0) (138) and lowest in the *Q. rugosa*: *P. durangensis* 40:60 and *Q. laeta*: *P. durangensis* 20:80 pellets (80). The corresponding value of the index in the *P. durangensis* pellets was 98.

The mechanical hardness of the pellets was negatively correlated with the amount of *P. durangensis* sawdust added (Fig. 4A). The hardness of the *Q. rugosa* pellets was most closely correlated with the amount of *P. durangensis* sawdust added ($r = -0.92$), while that of the *Q. conzattii* pellets was the least well correlated with the same parameter ($r = -0.09$). The IRI was also negatively correlated with the amount of *P. durangensis* sawdust added (Fig. 4B), and the correlation was most closely for *Q. sideroxyla* ($r = -0.95$).

The pellet bulk density was also negatively correlated with mechanical hardness and IRI (Fig. 5). The bulk density of *Q. sideroxyla* pellets most closely correlated with mechanical hardness ($r = -0.92$) and IRI ($r = -0.92$), while that of *Q. sideroxyla* pellets was least closely correlated with the same parameters ($r = -0.34$ and $r = -0.07$, respectively).

## Energetic properties

The calorific value did not differ significantly between species ($p = 0.24$), but there were significant differences between mixtures ($p = 1.82 \times 10^{-5}$) and for the species × mixture interaction ($p = 4.25 \times 10^{-6}$) (Fig. 6). The calorific value was highest in the 20:80 mixtures (above 19.7 MJ kg$^{-1}$). The values for the mixtures with the four oak species were in the range 19.0–19.8 MJ kg$^{-1}$. The calorific value of the *Pinus durangensis* pellets was highest (19.9 MJ kg$^{-1}$).

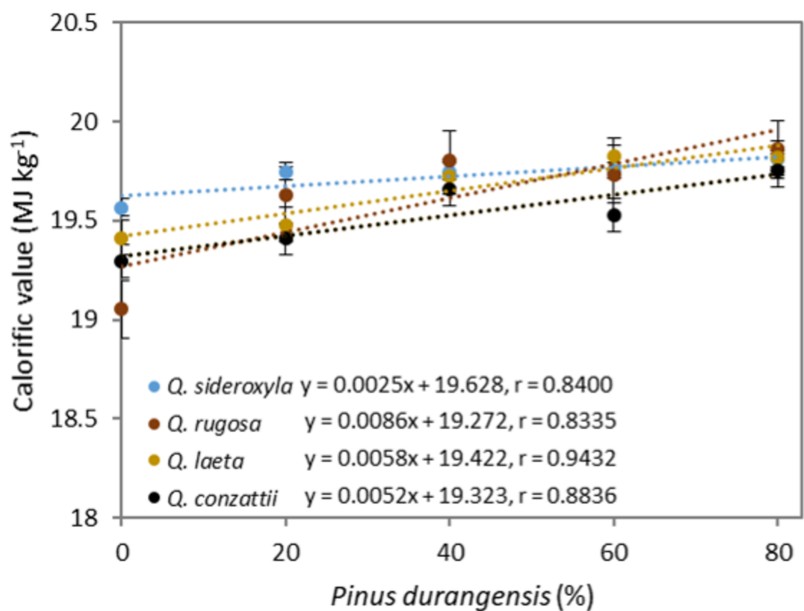

**Figure 6 Pellets calorific value correlated with the addition of *P. durangensis* sawdust.**

The calorific value was positively correlated with the amount of *P. durangensis* sawdust added. All values for the oak pellets were highly correlated with the amount of *P. durangensis* sawdust ($r > 0.83$).

## DISCUSSION

### Proximate analysis

The moisture content was below 10% for all species and mixtures, which are therefore classified as M10 class according to EN 14961-2 (*Spanish Association for Standarization (UNE), 2012*). Values for the oak-only pellets (100:0) were higher than for all mixtures with pine sawdust. This is because oak wood has fewer empty voids than pine wood, but large numbers of tyloses, which occlude the vessels and slow down the drying process (*De la Pérez-Olvera et al., 2015*). In general, the moisture content was below the 6.5% reported by *Zamorano et al. (2011)* for oak wood. It was lower than the range of 7.0–7.4% reported by *Miranda et al. (2011)* for mixtures of Pyrenean oak wood and washed grape pomace. Moisture content is a physical property that can be controlled by means of natural or artificial drying, and the values could therefore be optimized.

Volatile matter was lower in all oak-only pellets (100:0) than in the mixtures with pine sawdust. Values were similar to those reported by *Arranz et al. (2015)* for Pyrenean oak and pine forest residues (83.6 and 84.2%, respectively) and higher than those reported by *Miranda et al. (2011)* (range 67.8–83.6%). Values of volatile matter in pine were slightly lower than reported by *Rollinson & Williams (2016)* for pine pellets (83.6%) and by *Qin, Keefe & Daugaard (2018)*, who reported values in the range 81.2–82.6% for pellets from green beetle-killed and burned lodgepole pine. High volatile matter contents reveal important thermal properties. For example, pellet ignition is facilitated at low

temperatures, which increases the efficiency during the combustion process (*Torres-Ramos et al., 2015*). However, a low volatile matter content will hinder ignition of biofuel (*Vassilev, Vassileva & Vassilev, 2015*).

The ash content was generally lower in oak-only pellets than in the mixtures, except in the *Q. rugosa* (100:0) and *Q. laeta* (100:0) pellets. However, the ash contents of each species and mixtures were in accordance with the requirement specified by the standard EN- 14961-2 (*Spanish Association for Standarization (UNE), 2012*) for non-industrial use class A1 (≤0.7%). *Quercus rugosa* pellets were classified as A2 (≤1.5%), while *P. durangensis* pellets were classified as A1, and the values were similar to those reported by *Filbakk et al. (2011)* for *Pinus sylvestris* (0.47%).

The ash content was lower than the values reported by *Zamorano et al. (2011)* (3.3%) and by *Serrano et al. (2011)* (3.1%). *Herrera-Fernández et al. (2017)* mentioned that the ash content of the wood of some oak species reached 1.0%. Variations in ash content can be attributed to several factors, including physiological adaptations of the species (*Bárcenas-Pazos et al., 2008*), collection method, drying and handling of logs (*Zamorano et al., 2011*) and the proportion of bark in the wood (*Filbakk et al., 2011*; *Herrera-Fernández et al., 2017*; *Lerma-Arce, Oliver-Villanueva & Segura-Orenga, 2017*; *Núñez-Retana et al., 2019*). High ash contents cause slag formation, fouling and sintering (*Vega-Nieva et al., 2016*), negatively affecting the maintenance cost for both household and industrial users (*Carrillo-Parra et al., 2018*).

Lower values for fixed carbon were obtained for pellets made from each of the four oak species (100:0) than that reported by *Miranda et al. (2011)* (12.65%). The fixed carbon content of the pine pellets was similar to that reported by *Poddar et al. (2014)* (13.0%), and it was below the range of 16.9–18.43% reported by *Qin, Keefe & Daugaard (2018)*. Fixed carbon is an important bioenergy parameter due to its relationship with potential energy of solid fuels, and high fixed carbon contents are associated with high calorific power and with low moisture content and volatile matter content (*Chen, Peng & Bi, 2015*; *Forero-Nuñez, Jochum & Sierra, 2015*).

## Physical properties

The bulk density of the oak-only pellets (100–0) classified these as class BD550 (≥550 kg m$^{-3}$) according to standard EN 14961-1 (*Spanish Association for Standarization (UNE), 2011b*). These pellets are therefore of lower quality than pellets including pine sawdust, and classified as BD600, but complied with the specifications established by EN 14961-2 (*Spanish Association for Standarization (UNE), 2012*) (≥600 kg m$^{-3}$) for wood pellets destined for non-industrial use. The *P. durangensis* pellets were also classified as suitable for non-industrial use.

The bulk density was higher in the oak pellets containing pine sawdust than in those made from oak sawdust only (Fig. 3A). A similar response was observed by *Monedero, Portero & Lapuerta (2015)*. This can be explained by the higher bulk density of pine pellets relative to oak pellets.

Values of the bulk density of oak pellets were within the range reported for other species. For example, *Miranda et al. (2015)* reported bulk density values in the range

620–824 kg m$^{-3}$ for pellets made from different raw materials, and they were higher than those established by *Carrillo-Paniagua (2015)* for residues of *Hieronima alchorneoides* and *Eucalyptus* spp. (in the range 480–603 kg m$^{-3}$).

Bulk density is an important factor due to its relationship with the space required for storage and transport of the pellets (*Lehtikangas, 2001*; *Lerma-Arce, Oliver-Villanueva & Segura-Orenga, 2017*), as well as with the costs associated with these activities (*Zamorano et al., 2011*; *Garcia-Maraver et al., 2015*). Higher bulk density corresponds to more energy per unit volume, thus indicating greater economic benefits (*Rollinson & Williams, 2016*).

Particle density was lower in oak pellets (100:0) than in the oak-pine pellets, and the particle density was highest in the 80:20 mixtures. In a similar study carried out by *Liu et al. (2016)* with bamboo, the values were reduced by the addition of pine to the pellet mixture. In this study, the values increased in the (80:20) mixture and then decreased with the addition of *P. durangensis* sawdust until almost reaching the values obtained for the oak pellets (100:0). These researchers also reported values in the range 990–1,300 kg m$^{-3}$. These differences can be attributed to the relatively lower density of bamboo (540 kg m$^{-3}$) than of oak. However, the values obtained in the present study were higher than those reported by *Monedero, Portero & Lapuerta (2015)* (range 1,020–1,120 kg m$^{-3}$). The differences in density can be attributed to the fact that these researchers used poplar chips with particle density of 790 kg m$^{-3}$ in the mixtures, and in the present study the particle density was highest in the oaks, in the range 1,118–1,271 kg m$^{-3}$.

Similar values of pellet particle density were obtained by *García et al. (2019)* and *Bergström et al. (2008)*, who reported values in the range 1,259–1,276 kg m$^{-3}$. *Lehtikangas (2001)* reported values in the range 1,146–1,350 kg m$^{-3}$ for pellets made with different varieties of sawdust, logs and bark residues. *Jamradloedluk & Lertsatitthanakorn (2017)* reported particle density values in the range 1,300–1,800 kg m$^{-3}$, which is much higher than the values reported here. However, the differences may be due to the use of adhesives.

Particle density is an important parameter due to its influence on the apparent density and combustion behavior. Low density particles are needed in order to increase the burning period and energy production (*Qin, Keefe & Daugaard, 2018*). Particle density is also related to the moisture content of the raw material at the time of pelletizing, as at lower moisture content the friction increases through the plate in the matrix, which affects the movement of the particles and therefore, increases compression and density (*García et al., 2019*).

## Mechanical properties

The durability, expressed as the percentage mass retained after dropping the pellets twice, was within the range 67–78% for all oak pellets (100:0), that is, higher than the 51–74% obtained for the pellets containing *P. durangensis* sawdust. The lower durability of the pellets containing pine sawdust can be attributed to the very low abrasion index reported for this material (12%) (*Gil et al., 2010*).

*Abedi, Cheng & Dalai (2018)* used the drop test to determine the mechanical durability of spruce and oat hull pellets with additives (lignin and proline) and without additives. For pellets without additives, the durability ranged from 55% to 61%, and for pellets with additives, the values ranged from 60% to 90% (values similar to those reported here).

*Carrillo-Parra et al. (2018)* also observed significant differences ($p < 0.05$) in the retained mass in pellets made from three common tropical species. These researchers reported values of 49.4% for *Havardia pallens*, 61.7% for *Ebanopsis ebano*, and 66.2% for *Acacia wrightii*, respectively, without explaining the reason for these differences. These values were lower than reported here for oak species.

The impact resistance test, also known as the "drop resistance" or "shattering resistance" test (*Kaliyan & Morey, 2009*), enables estimation of the degree of compaction of the pellets and the resistance to breakage. These are important factors, as pellets must be able to support the transport, charge, discharge, storage and combustion processes to which they are subjected and will affect the efficiency of a pellet burner stove or burners (*Hu et al., 2015*). The impact resistance also enables evaluation of the mechanical durability through the shock and/or friction of densified fuels (*Temmerman et al., 2006*), as well as the strength of inter-particle bonds (*Forero-Nuñez, Jochum & Sierra, 2015*).

The values of the Impact Resistance Index (IRI) for the oak-only pellets (100:0) were in the range 113–138, and they were higher than in all mixtures, except for *Q. rugosa* 80:20 mixture and for the of *Q. laeta* and *Q. conzattii* 60:40 and 40:60 mixtures. Overall, the IRI was lower in the pellets containing *P. durangensis* sawdust. This can be attributed to the increase in the amount of *P. durangensis* sawdust which probably produced a pressure change in the pelletization process. This contrasts with the observations made by *Forero-Nuñez, Jochum & Sierra (2015)* for pellets made from cocoa shell mixtures and coal, in which the increase in cocoa shell mixture improved the impact resistance values, as with the use of fine particles (<0.297 mm). On the other hand, similar IRI values were reported by *Carrillo-Parra et al. (2018)* for *A. wrightii* (116 and 160).

The inverse correlation between the bulk density of the pellets and mechanical properties can be attributed to the higher bulk density as the pellets harden. Thus, when high-density pellets are dropped twice onto a concrete floor they will not absorb the impact and will break, while lower-density pellets are more likely to absorb the impact and not break. However, future studies should analyze the variations in shape and size distributions in relation to durability of bulk materials, in specific, bulk modulus and elastic response of pellets, as these factors are relatively poorly understood (*Wilson, 2010*).

The variations in the findings of several laboratories regarding the mechanical properties of pellets may be due to the different methods used to determine the characteristics of the pellets or the different devices used to produce the pellets. Further studies should be carried out to compare different methods used to determine the mechanical properties.

## Energetic properties

The calorific value of all pellets containing *P. durangensis* sawdust was slightly higher than that of the oak-only pellets. A similar pattern was described by *Serrano et al. (2011)*

because of the higher calorific value of pine. The values were above the limit established by standard EN 14961-2 (*Spanish Association for Standarization (UNE), 2012*) (16.5–19.0 MJ kg$^{-1}$), and the pellets can therefore be used for residential or industrial applications.

The calorific values obtained here were within the range reported for hardwoods (17.63–20.80 MJ kg$^{-1}$) by *Telmo & Lousada (2011)*. They are also similar to those reported by *Monedero, Portero & Lapuerta (2015)* for poplar and pine mixtures, and by *Miranda et al. (2011)* for Pyrenean oak waste. However, the addition of this raw material to washed grape pomace decreased the calorific value of the pellets. On the other hand, *Liu et al. (2016)* reported a value of 18.2 MJ kg$^{-1}$ for pine pellets, which is lower than the value reported here. The difference in values can be attributed to the physical and chemical properties, which can vary widely among different species (*Miranda et al., 2015*) and which are also influenced by the location, tree age, genetics and wood section in the canopy (*Dos Santos Viana et al., 2018*).

## CONCLUSIONS

The addition of *P. durangensis* sawdust to *Q. sideroxyla*, *Q. rugosa*, *Q. laeta* and *Q. conzattii* sawdust improved the bulk density and calorific value of the pellets made with the material. Making pellets with mixtures of oak and pine sawdust is therefore a potentially valuable alternative means of disposing of the by-products *Quercus* material generated by the forestry industry. On the other hand, the moisture and ash contents of the oak-pine pellets were in accordance with the limits established by standard EN 14961-2 (≤10% and ≤0.7%, respectively). Addition of the pine sawdust also improved the bulk density, with values reaching 703 kg m$^{-3}$, so that the pellets met the requirements specified by EN 14961-2 (≥600 kg m$^{-3}$). The mechanical hardness and IRI were lower in the pellets containing pine sawdust than in the other pellets. The calorific value of all mixtures increased with the addition of pine sawdust, reaching a maximum of 19.8 MJ kg$^{-1}$. Mixing oak and pine sawdust produced pellets with acceptable values for important traits included in the international standards, which are used as quality parameters.

## ACKNOWLEDGEMENTS

We thank Dr. Claudia Edith Bailón-Soto for assistance with the translation of this manuscript.

### Funding

This research was funded by Fondo de Sustentabilidad Energética, grant number SENER-CONACYT 2014 246911 Clúster de Biocombustibles Sólidos para la generación térmica y eléctrica y CONACYT project 166444. The funders had no role in study design, data collection and analysis, decision to publish, or preparation of the manuscript.

## Grant Disclosures

The following grant information was disclosed by the authors:

Fondo de Sustentabilidad Energética: SENER-CONACYT 2014 246911.

Clúster de Biocombustibles Sólidos para la generación térmica y eléctrica y CONACYT: 166444.

## Competing Interests

Christian Wehenkel is an Academic Editor for PeerJ.

## Author Contributions

- Víctor Daniel Núñez-Retana conceived and designed the experiments, performed the experiments, analyzed the data, prepared figures and/or tables, and approved the final draft.
- Rigoberto Rosales-Serna performed the experiments, authored or reviewed drafts of the paper, and approved the final draft.
- José Ángel Prieto-Ruíz analyzed the data, authored or reviewed drafts of the paper, and approved the final draft.
- Christian Wehenkel analyzed the data, prepared figures and/or tables, authored or reviewed drafts of the paper, and approved the final draft.
- Artemio Carrillo-Parra conceived and designed the experiments, performed the experiments, analyzed the data, authored or reviewed drafts of the paper, and approved the final draft.

## Field Study Permissions

The following information was supplied relating to field study approvals (i.e., approving body and any reference numbers):

The forest harvest permissions were approved and provided by the Secretariat of the Environment and Natural Resources on cutting areas Chinatú (SG.FO-08-2014/52), El Nopal (SG.FO-08-2014/129), El Pinto (SG.FO-08/2015/40), El Tule y Portugal (SG.FO-08-2014/82) and Llano Blanco (SG.FO-08/-2014/91) on Chihuahua State, Nicolas Romero (SG/130.2.2.2/002203/17) and El Regocijo (SG/130.2.2.2/002243/11) on Durango State.

## Data Availability

The raw data are available in the Supplemental File.

## Supplemental Information

Supplemental information for this article can be found online at http://dx.doi.org/10.7717/peerj.9766#supplemental-information.

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
