# Peer review of "Improving the physical, mechanical and energetic properties of Quercus spp. wood pellets by adding pine sawdust"

_PeerJ, doi:10.7717/peerj.9766_

## Round 0.1 · original submission · Major Revisions

Dear Dr. Carrillo-Parra,

Please see the attached reviews for detailed comments on your manuscript. Please revise or respond to their comments in detail. It is also suggested to consult a scientific editing service before resubmission.

Reviewer 1 ·

Basic reporting

The paper is clear with professional English language used throughout.
The introduction needs improvements mentioned in the general comments
The Literature review is good and relevant.
The structure is conform to PeerJ standards,
Figures are relevant, high quality, well labelled & described.
Raw data was supplied

Experimental design

The experimental design is correct and follows the standard analysis in this type of study

Validity of the findings

This paper doesn’t present significant novelty, but the results are acceptable. The conclusions are satisfactory. However, the justification for use this type of biomass (oaks) to pellet production is not sufficient.

Additional comments

The aim of this paper is to find better Quercus spp. pellets in their physical, mechanical and energetic properties by adding Pinus durangensis sawdust. The subject is interesting, but paper doesn’t present any novelty. Moreover, the paper mention that the Quercus is the second specie in Mexico, but is the supply (felling, thinning, etc) enough to motivate this study? In terms of Ecology, there’s is not a better utilization for this “residues”? The paper’s structure is good and the and analysis adequate, however, it needs better justification of the purpose of this study. Some detailed considerations are made below.

Detailed comments:
L 20 – “During logging, large amounts of waste are generated”. Please consider review this sentence. Nothing in forest is waste. The remain forest residues are nutrient for the ecosystem.
L 50-52 - “In addition, the use of biomass as fuel reduces greenhouse gas emissions during combustion and is considered neutral for carbon dioxide”. This sentence is controversy. Maybe the authors could include also some reference that claims that biomass is not neutral.
L 69 – “However, several species of Quercus genus which is the second most important forest resource “. Please provide the area occupied and the estimated residues generated by logging, thinning forest operations.
L183 – In order to be more comprehensive, consider write first the results and after comment the statistical results. Do the same in the other results
L183 – Please clarify in the text as well in the Methodology section that the MC is of the samples after “pelletisation” process.
L 185 …”lowest mean content”. Moister content?
L 232 – Please include here the full designation “impact resistance index (IRI)”

Reviewer 2 ·

Basic reporting

I suppose that overall quality of the manuscript is very low, for a relevant journal with a IF 2.35 as Journal of Life and Environmental Sciences is. First, I think that the 5.5 pages of references are too much exaggerated for the content of the article. The introduction should mention the potential for negative carbon dioxide budget for biochar from biomass(Line 52). In my view the Introduction is not very clear: e.g., (i) why the compostion in forest species change from logging methods lines, (lines 53 and 54) (ii) the densification and pelletizing (lines 60-67) are well matured in markets, and thus its description should be shorter and put in the beggining of the Introduction. The Introduction should also focus on the methodologies, for example in mechanical evaluation, used in M&M with same text which is in the Discussion. The obejctives (lines 95-97) with only three lines should be more detailed. The english phrasing should also be improved: for example "narrow" in line 71 or "reach international standarts" in line 78 are not clear wordings or the Introduction has too many paragraphs for its content.

Experimental design

If the elementary analysis (C,H,N,O,S) was not possible, and only proximate analysis was, a justification should be given, or indeed better that analysys, which is very common for biomass characteerization should be included; Wood or composite hardness should cahacterized through texts with stress values in SI units (e.g. Janka hardness test) under a more precies methodology than drop test is.

Validity of the findings

In the Results section p-values should not given rates of e.g. 4.09*10^(-5). P-values lower than say 0.001, should be enough. Moreover the results should be given in Tables and not only in graphs (Figures 2-5). Besides, some photos of pellet specimens with pine sawdust should be added. In the discussion moisture value should not given the elaboration shown insofar that moisture is controlled by drying(artificial or natural) and thus its values are transient. Oppositelly the eleboration given to vlolatiles and fixed carbon in discussion is very short and should include same considerations related to the structure of the materials. The ash discussion is also too long as it isand should directed to the relations with final use of pelletes: domestic or industrial. The discussion on bulk density is also too lenghty and correlations between bulk density and mechanical beheviour are missing and should of could also be included. In mechanical properties discussion the limitations of drop test should justify a comparision with e.g. Janka hardeness. Also, the absence of Tables turns the presentation of results and discussion as inadequate. Also I don't understand how a decrease in IRI values (line 387) improves mechanical hardness (line 420) in the Conclusions. Also an evaluation of the interconectedness between bulk density and calorific power should be included, e.g. though correlation analysis, of variables such fuelwood value index or energy density. This would contribute to improve the coherence of the text.
So overall I suppose that this work of improving pellet quality through pine sawdust addtion has potential but the narrative carried out is not enough to demonstrate that potential and I would recommend a major revision for a further review.

---

## Round 0.2 · Minor Revisions

Please include standard error represented by error bars on your plotted values. It also would help to have an English scientific editor edit your manuscript. Thank you for making significant improvements to your initial submission to PeerJ.

Reviewer 1 ·

Basic reporting

no comment

Experimental design

The experimental design was fair despite the proximate analysis of pellets (important parameter) were nor determined.

Validity of the findings

no comment

Additional comments

I was carefully reading the paper improvements and the authors answered the reviewer’s questions and the paper was significantly improved. Nevertheless, my question about the soundness and global quality of this papers remains. The paper has a low average quality for the high standard quality of PeerJ. However, if the decision is to publish, I do not detect any major inconsistencies in the methodology and results presented.

---

## Round 0.3 · accepted · Accept

Dear Dr. Carrillo-Parra,
Thank you for resubmitting your manuscript in a timely manner and making the suggested corrections. We hope you will consider PeerJ for future manuscripts detailing your work.

Best Regards,
Scott Wallen